# Trends in Gynecologic Carcinosarcoma Based on Analysis of the Surveillance Epidemiology End Result (SEER) Database

**DOI:** 10.3390/jcm12031188

**Published:** 2023-02-02

**Authors:** Joo Won Lee, Yung-Taek Ouh, Ha Kyun Chang, Kyung-Jin Min, Sanghoon Lee, Jin-Hwa Hong, Jae-Yun Song, Jae-Kwan Lee, Nak Woo Lee

**Affiliations:** 1Department of Obstetrics and Gynecology, Korea University Ansan Hospital, 123, Jeokgeum-ro, Danwon-gu, Ansan-si 15355, Gyeonggi-do, Republic of Korea; 2Department of Obstetrics and Gynecology, School of Medicine, Kangwon National University, 156, Baengnyeong-ro, Chuncheon-si 24289, Gangwon-do, Republic of Korea; 3Department of Obstetrics and Gynecology, Korea University Anam Hospital, 73, Goryeodae-ro, Seongbuk-gu, Seoul 02841, Republic of Korea; 4Department of Obstetrics and Gynecology, Korea University Guro Hospital, 148, Gurodong-ro, Guro-gu, Seoul 08308, Republic of Korea

**Keywords:** carcinosarcoma, SEER, uterine, ovarian, cervical, trends, prognosis, survival

## Abstract

Carcinosarcomas (malignant mixed Mullerian tumors) of a female genital organ are rare tumors associated with a poor survival. The purpose of this study was to identify site-specific differences in the incidence and prognosis in carcinosarcomas originating in the uterus, cervix, or ovary. The data of patients with gynecologic carcinosarcomas were extracted from the Surveillance, Epidemiology, and End Results (SEER) database between 2000 and 2016. The characteristics of gynecologic carcinosarcomas were compared using Pearson X2 and Fisher’s exact tests. Kaplan–Meier models were used for cause-specific survival (CSS) analysis. The cohort included 7086 females, including 5731 cases of uterine carcinosarcoma, 161 cervical carcinosarcomas, and 1193 ovarian carcinosarcomas. The age-adjusted incidence rates of uterine, cervical, and ovarian carcinosarcoma were 3.9, 0.1, and 0.6 per 1,000,000, respectively. In the distribution of carcinosarcoma incidence by race, compared with the uterus or cervix, those originating from the ovary were unequally distributed in Caucasians (84.4% versus 69.6%, 67.7%; *p* < 0.001). The incidence of uterine carcinosarcoma steadily increased over time, from 2.2 in 2000 to 5.5 in 2016 (per 1,000,000), while cervical or ovarian carcinosarcoma showed no significant difference in incidence. The five-year CSS rates based on the site of origin (uterus, cervix, and ovary) were 39.9%, 33.1%, and 25.8%, respectively. The incidence rates of gynecologic carcinosarcoma, especially uterine carcinosarcoma, are gradually increasing. Although uterine carcinosarcoma is associated with a higher incidence than the others, it has a better prognosis compared with ovarian and cervical carcinosarcoma. The survival rates were worst in ovarian carcinosarcoma.

## 1. Introduction

Carcinosarcomas, also known as malignant mixed Müllerian tumors, are rare but highly aggressive epithelial malignancies that contain both malignant sarcomatous and carcinomatous elements [1]. Gynecologic carcinosarcoma mostly originate in the uterus, yet uterine carcinosarcoma accounts for less than 5% of all uterine malignancies [2]. Compared to other high-risk malignancies of a similar stage, uterine carcinosarcoma has a rapid progression and poor prognosis [3]. Ovarian carcinosarcoma is the second most common gynecologic carcinoma, constituting 1–3% of ovarian carcinoma [4]. Regardless of the primary site, the prognosis for gynecologic carcinosarcoma is poor, with an overall 5-year survival rate of less than 30%.

Uterine carcinosarcomas are uncommon and comprise less than 5% of uterine tumors, with more than 35% of cases presenting with extrauterine disease at diagnosis [3]. Uterine carcinosarcoma has been traditionally included in the sarcoma category and was the most common uterine sarcoma [5]. However, it has recently been classified as a high-grade endometrial cancer [6,7]. Uterine carcinosarcoma accounts for about 2–5% of endometrial cancers [3]. Precursor lesions of complex hyperplasia with atypia are found in more than 40% of cases. Uterine carcinosarcoma accounts for 15% of all deaths caused by malignant tumors of uterine corpus, although it accounts for less than 5% of all uterine malignant tumors [8]. Ovarian carcinosarcomas represent less than 1% of all ovarian tumors, and once diagnosed, about 90% of cases will experience metastasis. Several observational studies have suggested that ovarian carcinosarcomas follow a distinct natural history compared with other more common epithelial carcinomas [4]. In addition, cervical carcinosarcomas are the rarest malignancies associated with Müllerian ducts and mesonephric duct remnants. Owing to the relative infrequency of the disease, most of the available data pertaining to the natural history of cervical carcinosarcomas are derived from case reports and small case series [9].

The recognition of trends and changes in the characteristics of carcinosarcomas, as well as the prevalence, recurrence, and poor overall survival, is useful in understanding the epidemiology of gynecologic carcinosarcomas. Nevertheless, published data regarding the trends in incidence and prognosis are not enough. The objective of this study is to investigate the site-specific differences in the incidence and prognosis by comparing carcinosarcomas originating in the uterus, cervix, and ovary.

## 2. Materials and Methods

### 2.1. Patients

Data involving patients with gynecologic carcinosarcomas were extracted from the Surveillance, Epidemiology, and End Results (SEER) database from 2000 to 2016; this database collects patients’ clinicopathological features and survival data. The SEER*Stat 8.3.6 (IMS Inc., Calverton, MD, USA) was used to extract the SEER18 dataset, which contains reliable information on cancer incidence and survival derived from 18 population-based cancer registries, covering approximately 34.6% of the US population [10]. Patients with a primary diagnosis of cancer of the uterus, cervix, or ovary were identified to represent the study cohort, using the SEER International Classification of Disease for Oncology (ICD-O-3) codes based on histological subtypes: “Carcinosarcoma, NOS (8980/3)” with “uterine cancer (C54, C55), cervical cancer (C53), or ovarian cancer (C56).”

### 2.2. Clinicopathological Characteristics

The following information was obtained to investigate the clinicopathological features of gynecologic carcinosarcoma: age, race (White, Black, American Indian/Alaska Native, Asian or Pacific Islander, and others), tumor histology, grade (well differentiated, moderately differentiated, poorly differentiated, undifferentiated, and unknown), and the type of treatment received (surgery, radiation, both, or unknown/no treatment). The tumor stage was assessed according to the SEER historic staging system (localized, regional, distant, and unknown) for consistent cancer staging over all the research periods. The incidence rates were calculated per 1,000,000 person-years and age-adjusted to the 2000 US standard population. The age-adjusted incidence rates with 95% confidence intervals were analyzed by primary site, race, historic stage, age group (20–29, 30–39, 40–49, 50–59, 60–69, 70–79, and 80+ years), and the lymph nodes assessed. The cause-specific survival (CSS) rate was calculated at 5 years.

### 2.3. Statistical Analyses

All continuous variables were analyzed using Student’s *t*-test and analysis of variance (ANOVA). Comparisons of the characteristics between gynecologic carcinosarcomas were performed using Pearson X2 and Fisher’s exact tests. CSS was defined as the time from the date of diagnosis to the date of death from carcinosarcoma. Overall survival (OS) was defined as the time from the date of diagnosis to the date of death due to any cause or to the last follow-up. The incidence rates were calculated and exported to IBM Statistical Product and Service Solutions (SPSS©), version 20.2 (IBM Corp., Armonk, NY, USA). Kaplan–Meier models were used for CSS analysis. Statistical significance was accepted at *p* < 0.05.

### 2.4. Ethics Approval and Consent to Participate

All procedures involving human participants followed the 1964 Declaration of Helsinki and subsequent amendments or the equivalent ethical standards. We signed the ‘Surveillance, Epidemiology, and End Results Program Data Use Agreement’ according to the database’s use requirements. Approval was waived by the local institutional review board because SEER data are public and anonymized.

## 3. Results

From 2000 to 2016, the total cohort included 7086 cases of gynecologic carcinosarcoma, including 5731 cases of uterine carcinosarcoma, 161 cervical carcinosarcomas, and 1193 ovarian carcinosarcomas. The clinical and demographic characteristics of the cohort are presented in Table 1.

Patients with uterine carcinosarcoma were older compared to those with cervical or ovarian cancer. The mean age at diagnosis was 67.7 ± 11.3 years. Gynecologic carcinosarcomas were most prevalent in patients in their 60s (34.8% of uterus, 32.3% of cervix, and 28.2% of ovary). They were prevalent in White women, and gynecologic carcinosarcoma predominance in that racial group is shown by case incidence: ovary, 84.4%; uterus, 69.6%; and cervix, 67.7% (*p* < 0.001). The distribution of stage at diagnosis varied depending on the primary site of the carcinosarcoma. In uterine carcinosarcoma, the localized stage was the most prevalent (37.6%), and the distant stage was found in only 23.4% of cases. Conversely, in ovarian carcinosarcoma, the distant stage was the most prevalent (72.3%), while the localized stage was observed in only 5.2% of patients (*p* < 0.001). In the carcinosarcoma of the cervix, the regional stage was the most common (43.5%), followed by distant and localized stages accounting for 29.2% and 22.4%, respectively (*p* < 0.001). Most patients (96.5%) diagnosed with ovarian carcinosarcoma did not undergo radiotherapy, whereas patients with other gynecologic carcinosarcomas more frequently underwent adjuvant radiotherapy (uterus, 32.8%; cervix 25.4%, *p* < 0.001). In terms of sequence of radiotherapy, adjuvant radiotherapy was most frequent (uterus, 32.2%; cervix 21.1%). External beam radiotherapy (EBRT) was the most common treatment method (19.8% for uterus; 29.8% for cervix, *p* < 0.001). Vaginal brachytherapy (VBT) was performed more frequently for carcinosarcoma, other than ovarian carcinosarcoma (uterus, 15.8%; cervix, 14.3%). While 73.1% of ovarian carcinosarcomas were treated with chemotherapy, only 39.8% of cervical carcinosarcomas were managed with chemotherapy (*p* < 0.001). Lymph nodes were examined in 65.5% of uterine carcinosarcomas, and positive lymph nodes were found in 18.9%. By contrast, only 37.3% of cervical cancers had invaded lymph nodes (*p* < 0.001).

The age-adjusted incidence rates of carcinosarcoma (per 1,000,000) involving the uterus, cervix, or ovary were 3.9, 0.1, and 0.8, respectively, from 2000 through 2016 (Figure 1). The incidence of uterine carcinosarcoma steadily increased, from 2.2 in 2000 to 5.5 in 2016 (per 1,000,000), while other gynecologic carcinosarcomas showed no significant difference in incidence over the selected time range.

The five-year CSS rates according to sites of origin, including the uterus, cervix, and ovary, were 39.9%, 33.1%, and 25.8%, respectively (Figure 2). The 1-year survival rate was lowest for patients with cervical carcinosarcoma (cervix, 55.1%; uterus, 70.5%; ovary, 62.3%), whereas the 5-year survival rate of those with ovarian carcinosarcoma was lowest (ovary, 25.8%; uterus, 39.9%; cervix, 33.1%). Uterine carcinosarcoma showed the best prognosis compared with other gynecologic carcinosarcoma (*p* < 0.001).

## 4. Discussion

This study presented the trends in incidence and CSS rates of gynecologic carcinosarcoma in the U.S. from 2000 through 2016. The incidence of uterine carcinosarcoma has increased over the specified timeframe in the U.S., a trend not observed in other gynecologic carcinosarcomas. Patients with carcinosarcoma of the uterus showed better 5-year disease-specific survival rates, whereas ovarian carcinomas were associated with a poor prognosis.

Gynecologic carcinosarcoma was most prevalent in women in their 60s, consistent with a population-based study in Israel [11]. During the last decades, the number of women older than 60 years has increased worldwide with the trend of an increasing life span. As a consequence, the focus in older women will be on lifestyle support to counteract various degenerative features. In addition, studies investigating gynecologic disease will be increasingly in demand.

In our study, the distribution of carcinosarcoma according to primary organ varied slightly by race. Compared with cervical carcinosarcoma, ovarian carcinosarcoma involved a higher percentage of White women, whereas the percentage of Black women was lower (cervix: 67.7% in White, 24.2% in Black; ovary: 84.4% in White, 8.2% in Black). These incidental differences in race could be attributed to genetic factors, cultural differences, and socioeconomic status. A previous study showed racial disparities in disease characteristics and survival in gynecologic carcinosarcoma [12]. In a U.S. study, the characteristics, treatment, and survival of patients with uterine and ovarian carcinosarcoma were compared between Black women and White women. Black women diagnosed with both uterine and ovarian carcinomas showed a worse overall survival [13]. A comparison of Jewish and non-Jewish women revealed significantly higher incidence in the Jewish women [11]. Unfortunately, there are insufficient data on the prevalence or prognosis of carcinosarcoma in Asians. According to a study conducted in Japan [14], it was very rare in the cervix or ovary, but in endometrial cancer, it has been investigated as the second most common histological type following endometrioid adenocarcinoma (5.2%). Nevertheless, data on the trends or prognosis of gynecologic carcinosarcoma over time are lacking.

Gynecologic carcinosarcomas are rapid growing tumors with prevalence and recurrence rates comparable to common gynecologic malignancies. The 5-year disease-specific survival rate associated with ovarian carcinosarcoma was inversely correlated with the disease stage [15,16]. Similar results have been reported for uterine carcinosarcoma, which adversely affects survival in advanced stages regardless of adjuvant therapy [17,18]. Our results indicate better 5-year survival rates in uterine carcinosarcoma, followed by cervical and ovarian carcinosarcoma.

Cervical carcinosarcoma was reported to occur in 0.2 per 100,000 people in Europe [19]. It has a better prognosis than carcinosarcoma from the uterus, ovary, or other organs because it is likely to be detected relatively early due to symptoms such as vaginal bleeding [20]. Similarly to squamous cell carcinoma or adenocarcinoma commonly occurring in the cervix, cervical carcinosarcoma is also associated with HPV infection, but the exact role of HPV needs further research. According to a previous literature, all eight cases of cervical carcinosarcoma were associated with high-risk HPV infection [21]. The widespread introduction of the HPV vaccine has reduced cervical carcinoma [22], but there was no data on whether the incidence of carcinosarcoma has decreased, so further research is needed.

The types of adjuvant treatment according to the primary site of carcinosarcoma also differed slightly in our study. The optimal treatment for gynecologic carcinosarcoma is controversial. Adjuvant therapies for uterine carcinosarcoma have advanced over time. Compared with locally advanced or metastatic disease, uterine carcinosarcoma can be treated via total abdominal hysterectomy and bilateral salpingo-oophorectomy with or without pelvic and para-aortic lymphadenectomy. The decision to administer adjuvant treatments, including pelvic external beam radiotherapy, chemotherapy, and hormonal therapy, depends on the presence of certain risk factors. The benefits of adjuvant chemotherapy and radiation therapy for patients with uterine cancer carcinosarcoma were reported by the Gynecologic Oncology Group 150 [23]. However, recent data suggested that adjuvant chemoradiotherapy was superior to chemotherapy alone in uterine cancer carcinosarcoma [24]. Although prospective data are lacking, the management of ovarian carcinosarcomas is similar to that of other ovarian tumors and typically entails cytoreductive surgery, followed by adjuvant chemotherapy for women with an advanced stage disease [4]. Due to the lack of information regarding cervical carcinosarcoma, no consensus is available regarding disease prognosis and treatment [9].

Previous studies [25,26] reported an increase in the incidence of gynecologic carcinosarcoma, but no individual carcinomas were analyzed. In our present study, as discussed above, an increase in the incidence of overall gynecologic carcinosarcoma was found, but the increase in uterine carcinosarcoma was particularly notable. Uterine carcinosarcoma and endometrial adenocarcinomas share several similar risk factors. The risk of these two diseases increases with an elevated estrogen level and decreases with a history of oral contraceptive use. Obesity and nulliparity were other risk factors. Nevertheless, compared with Grade 3 endometrial cancer, women diagnosed with uterine carcinosarcoma are older, with a median age of 70 years. Therefore, the incidence of uterine carcinosarcoma has showed a significantly greater increase than cervical and ovarian carcinosarcomas, which is consistent with the results reported in previous studies [25,26].

The study has several strengths, including the large number of patients with gynecologic carcinosarcoma derived from the SEER database, which accurately reflects the incidence and survival rates of rare carcinosarcoma. Nevertheless, this study has limitations. Since many cases involved missing TNM stages, an accurate summary stage analysis could not be performed. Second, survival analysis was not performed specifically by race, age, or stage, nor was there a trend analysis of survival. Although survival analysis involved the primary sites of the carcinosarcoma, no survival analysis by stage was performed.

## 5. Conclusions

In conclusion, the incidence rates of gynecologic carcinosarcoma, especially uterine carcinosarcoma, have gradually increased. Although the incidence of uterine carcinosarcoma is higher than that of the others, it is associated with a better prognosis compared with cervical and ovarian carcinosarcomas. The high recurrence rate and poor overall survival of gynecologic carcinosarcoma suggest the need for improved management strategies. Given the rarity of carcinosarcoma, however, large international prospective trials are warranted to establish treatment regimens.

## Figures and Tables

**Figure 1 jcm-12-01188-f001:**
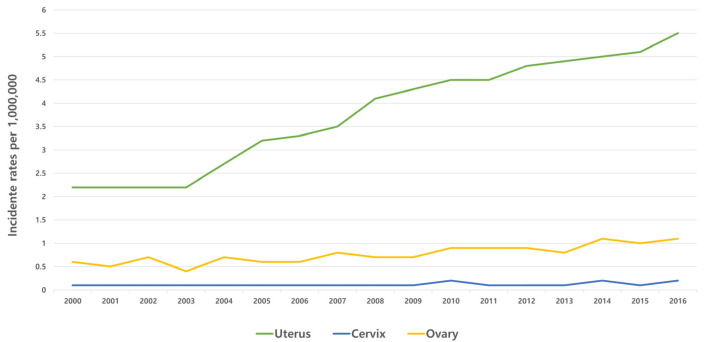
The annual trends in the incidence of gynecologic carcinosarcoma (2000 to 2016).

**Figure 2 jcm-12-01188-f002:**
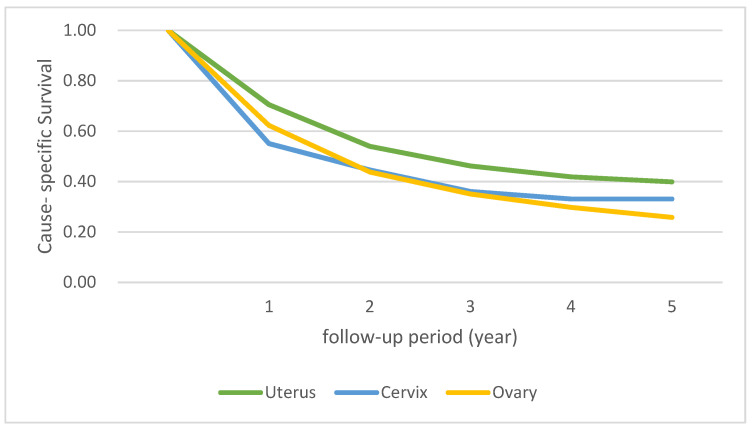
Comparison of cause-specific survival rates according to origin of gynecologic carcinosarcoma.

**Table 1 jcm-12-01188-t001:** Demographic characteristics of gynecologic carcinosarcoma.

Primary Site	Uterus	Cervix	Ovary	*p*-Values
No. of patients	5731	161	1193	
Age (year)				<0.001
Mean ± SD	67.7 ± 11.3	66.1 ± 13.8	66.3 ± 12.1	
20–29	10 (0.2)	2 (1.2)	5 (0.4)	
30–39	54 (0.9)	2 (1.2)	17 (1.4)	
40–49	225 (3.9)	11 (6.8)	73 (6.1)	
50–59	985 (17.2)	32 (19.9)	258 (21.6)	
60–69	1994 (34.8)	52 (32.3)	336 (28.2)	
70–79	1550 (27.0)	29 (18.0)	322 (27.0)	
80<	913 (15.9)	33 (20.5)	182 (15.3)	
Marital Status				<0.001
Married	2462 (43.0)	60 (37.3)	577 (48.4)	
Never married	954 (16.6)	30 (18.6)	198 (16.6)	
Separated	665 (11.6)	24 (14.9)	141 (11.8)	
Widowed	268 (4.7)	33 (20.5)	234 (19.6)	
Unknown	1382 (24.1)	14 (8.7)	43 (3.6)	
Race				<0.001
Caucasian	3989 (69.6)	109 (67.7)	1007 (84.4)	
Black	1297 (22.6)	39 (24.2)	98 (8.2)	
American Indian/Alaska Native	28 (0.5)	1 (0.6)	7 (0.6)	
Asian or Pacific Islander	404 (7.0)	11 (6.8)	75 (6.3)	
Unknown	13 (0.2)	1 (0.6)	6 (0.5)	
Grade				<0.001
Well-differentiated; Grade 1	58 (1.0)	1 (0.6)	3 (0.3)	
Moderately differentiated; Grade 2	141 (2.5)	1 (0.6)	12 (1.0)	
Poorly differentiated; Grade 3	2206 (38.5)	65 (40.4)	380 (31.9)	
Undifferentiated; anaplastic; Grade 4	1275 (22.2)	25 (15.5)	313 (26.2)	
Unknown	2051 (35.8)	69 (42.9)	485 (40.7)	
Summarized stage				<0.001
Localized	2153 (37.6)	36 (22.4)	62 (5.2)	
Regional	2077 (36.2)	70 (43.5)	256 (21.5)	
Distant	1341 (23.4)	47 (29.2)	862 (72.3)	
Unknown	160 (2.8)	8 (5.0)	13 (1.1)	
Sequence of radiotherapy				<0.001
Adjuvant radiotherapy	1848 (32.2)	34 (21.1)	37 (3.1)	
Neoadjuvant radiotherapy	35 (0.6)	7 (4.3)	3 (0.3)	
None	3823 (66.7)	120 (74.5)	1151 (96.5)	
Others	25 (0.4)	0 (0.0)	2 (0.2)	
Radiotherapy				<0.001
EBRT	1132 (19.8%)	48 (29.8)	42 (3.5)	
EBRT + VBT	376 (6.6)	15 (9.3)	0 (0.0)	
VBT	525 (9.2)	8 (5.0)	1 (0.1)	
None	3426 (59.8)	78 (48.4)	1136 (95.2)	
Others	22 (0.4)	1 (0.6)	1 (0.1)	
Refused	87 (1.5)	7 (4.3)	1 (0.1)	
Unknown	163 (2.8)	4 (2.5)	12 (1.0)	
Chemotherapy				<0.001
Yes	2851 (49.7)	64 (39.8)	872 (73.1)	
None/unknown	2880 (50.3)	97 (60.2)	321 (26.9)	
Regional examined nodes				<0.001
Examined nodes	3754 (65.5)	60 (37.3)	519 (43.5)	
Positive	1086 (18.9)	20 (12.4)	176 (14.8)	
Negative	2668 (46.6)	40 (24.8)	343 (28.8)	
No examined nodes	1977 (34.5)	101 (62.7)	674 (56.5)	

Values are presented as number (%). SD, standard deviation; EBRT, external beam radiotherapy; VBT, vaginal brachytherapy.

## Data Availability

The data analyzed in this study is available on the SEER database (http://seer.cancer.gov/, accessed on 25 January 2023).

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
