# Peer review of "Trends in Gynecologic Carcinosarcoma Based on Analysis of the Surveillance Epidemiology End Result (SEER) Database"

_jcm, 2023, doi:10.3390/jcm12031188_

Round 1

Reviewer 1 Report

Dear authors,

Complements on your analysis of SEER data on various MMMTs.

1. What is the main question addressed by the research?

Any population database analysis does give us an insight about the 'trends' of a given disease, and provide an epidemiological insight. However like any other retrospective study, the drawbacks are inherent in this manuscript as well. And that it represents only a certain geographic region. The study adds to existing information though I am concerned about collating uterine, cervical and ovarian Carcinomas in one study. Though they are all gynecological organs, the disease behaviour is different for all 3. Cervical & Ovarian carcinomas have a worse outcome compared to ovary, which is highlighted in this study as well. This study is not entirely novel but strengthens the knowledge that Carcinosarcoma is a poor-biology disease irrespective of origin.

I feel all three diseases sites should be separately described and a joint conclusion avoided.

The cited references are appropriate.

Reviewer 2 Report

The authors have submitted a short review article on gynecologic carcinosarcomas using the SEER database in the United States. The authors show that carcinosarcomas of the uterine corpus have been on the rise in recent years, and that ovarian carcinosarcomas have a poor prognosis. The manuscript is generally well described, and this reviewer would like to suggest several potential improvements.

#1. Figure 2 compares cause-specific survival in the three groups of patients (uterus, cervix, and ovary), however the prognosis should be better compared by survival time analysis, as there should be a mix of cases with different follow-up periods. It would be better to illustrate the KM curves of the three groups for comparison.

#2. As for cervical carcinosarcomas, most of them are considered to be HPV-associated tumors. Could you comment in Discussion on the relationship between HPV vaccination in the U.S. and the subsequent frequency of cervical carcinosarcoma?

#3. Although the authors are a Korean research group, there are relatively few data on Asian patients in the SEER database. Are there any racial differences in the two important findings of this study, the increasing trend of uterine carcinosarcoma and the poor prognosis of ovarian carcinosarcoma? Could you add to the discussion by citing data from, for example, Korea, Japan, and China, to see if there is a similar trend in Asians. Those considerations would be of help to readers.

Round 2

Reviewer 1 Report

Dear authors

The manuscript is much improved now

Reviewer 2 Report

The authors made adequate revisions according to the comments and the issues have been clarified.